# The Unique Role of Hope and Optimism in the Relationship between Environmental Quality and Life Satisfaction during COVID-19 Pandemic

**DOI:** 10.3390/ijerph19137661

**Published:** 2022-06-23

**Authors:** Walton Wider, Nasehah Mohd Taib, Mohd Wafiy Akmal Bin Ahmad Khadri, Foon Yee Yip, Surianti Lajuma, Prasath A/L Punniamoorthy

**Affiliations:** Faculty of Business and Communication, INTI International University, Nilai 71800, Malaysia; nasehah.taib@newinti.edu.my (N.M.T.); wafiyakmal.khadri@newinti.edu.my (M.W.A.B.A.K.); foonyee.yip@newinti.edu.my (F.Y.Y.); surianti.lajuma@newinti.edu.my (S.L.); prasathprazanthe@gmail.com (P.A/L.P.)

**Keywords:** environmental quality, life satisfaction, hope, optimism, private university students

## Abstract

COVID-19 in Malaysia has significantly affected the higher education system of the country and increased the level of distress among university students. Empirical evidence proposed that environment quality is associated with university students’ life satisfaction during COVID-19. It was found that hope and optimism are linked with greater life satisfaction in general. Although past literature has reported the effects of hope and optimism on life satisfaction, there are limited studies examining the underlying mechanism among Malaysian private university students. Therefore, the current study offers the preliminary understanding of the intervening role of hope and optimism on the relationship between environmental quality and life satisfaction among private university students in Malaysia. A total of 133 private university students in Malaysia were recruited through homogenous convenience sampling. Partial least square structure equation modeling (SmartPLS) was used to analyze the mediation models. The results revealed that only hope mediated the relationship between environmental quality and life satisfaction, but not optimism. Hence, it is proposed that mental health providers should focus on providing hope-related interventions to university students in confronting COVID-19 challenges and ultimately improving life satisfaction.

## 1. Introduction

Due to the emergence of the COVID-19 pandemic, most of the higher education institutions have opted for fully online classes for their students as it was risky and impractical for them to undergo the usual face-to-face classes [1]. Ever since, many countries, for example, Malaysia being one of them, started implementing online classes instead of the traditional classes, just to safeguard the wellbeing and wellness of their students [2]. Though online studies have brought various advantages to students in Malaysia on the whole, there are still certain disadvantages to it [3]. Since attending online class, students are reported to have gone through a higher rate of anxiety and depression [2]. Moreover, students were reported to experience severe social dysfunction and drop in their confidence level [4] due to lack of knowledge and crisis management [5]. In Malaysia, studies reported that the main issues for students are internet connectivity, ineffective online learning method, slow personal laptops and gadgets, and difficulty concentrating during online learning [6]. It is additionally appalling to note that students attended online classes for six to eight hours daily by using their mobile phones, which further added to unrealistic stress and medical issues [7]. In addition, having difficulty in communication with lecturers, poor engagement with peers, and lack of access to laboratory significantly impacted the learning experience of students [8].

The emergence of the COVID-19 pandemic leads to people getting more concerned about life satisfaction, particularly among students [9]. Before the pandemic, environmental factors have been reported to play a significant role in promoting life satisfaction [10,11,12,13]. In the university context, in order to help students to reduce stress and build a sense of spirit, they are encouraged to participate in the activities organized by clubs and societies [14]. It is reported that participating in a variety of university activities will help students pursue a positive well-being [15]. Therefore, universities should encourage students to be active in campus activities by providing a healthy environment for students to have opportunities to achieve psychological well-being [15]. However, after the pandemic struck, there is even greater awareness of the role environmental quality plays in life satisfaction [16]. In order to deal with COVID-19, which has put citizens’ physical and mental health in jeopardy, the government has imposed a series of mandatory isolation measures [17]. As a result, residents were constrained within their local neighbourhoods. Specifically, empirical evidence showed a significant linkage between environmental quality and life satisfaction during COVID-19 among university students in Malaysia. According to Ref. [18], students staying outside the campus showed higher life satisfaction than those who were confined to the house or hostel. The freedom and flexibility of having to engage with many activities during the lockdown explained why students staying outside the campus house are more satisfied with their lives. Given these circumstances, there is a necessity to have a closer observation into the environmental quality of students’ lives amidst COVID-19 as the state is closely related to life satisfaction. Furthermore, Bakkeli [19] stated the need to identify the underlying mechanism of students’ life satisfaction during the COVID-19 pandemic. Additionally, study on life satisfaction from the perspective of positive psychology and environmental psychology is warranted amidst COVID-19 [20] now more than before, as cultivating hope and optimism become more revelatory in measuring the quality of environment and satisfaction of life in times of the COVID-19 pandemic [21]. Hope and optimism act as cognitive factors in protecting and helping to reduce life stressors in students’ lives [22], particularly life satisfaction [23]. Furthermore, ref. [21] asserted that optimism and hope can be considered as the fundamental paths in adapting to horrendous life occasions by trusting in a greater future, and along these lines, may act as interceding factors. It is well documented in the literature that hope and optimism are consider two distinct concepts [24]. In many ways, hope and optimism can still be similar in a way that both hopeful and optimistic individuals are looking forward to a positive outcome to happen in the future [25]. Theories that were used as the foundation of our research framework include the Hope Theory [26] and Optimism Theory [27]. These studies are the first preliminary studies that integrate environmental quality, hope, optimism, and life satisfaction amidst COVID-19 among Malaysian private university students. Therefore, this preliminary study would increase the diversity of this theory, especially in the Malaysian context. Motivated by a similar reason, this study was conducted in the hope of contributing additional empirical data to the development of both theories.

The crucial question is how does hope and optimism play an intervening role in a students’ life satisfaction? Is it better for students to be more hopeful or optimistic amidst COVID-19? Therefore, the current study is aimed at answering the question by examining the intervening role of hope and optimism on the relationship between environmental and life satisfaction among private university students in Malaysia. It is hoped that the findings of this study could help mental health providers to lay out practical skills to promote life satisfaction among university students in challenging situations such as COVID-19.

### Hope and Optimism as Mediators

Hope includes two interrelated cognitive processes such as pathways and agency [28]. Therefore, hope can be characterized as the perceived ability to infer pathways to desired goals and one’s capacity to motivate oneself through agency thinking to utilize those pathways [29]. Thus, people who hold onto hope are usually those candidates who own an immense desire to get their achievements unlocked diligently [30]. One of the reasons why hopeful people have an immense amount of motivation in pursuing their goals is they generally use hope as a fuel to provide them strength, especially when it comes to stress management [31]. On the other hand, optimism is often expressed as one’s ability of anticipating desired outcomes of future activities [32]. Aided by certain coping strategies in life, optimistic people are able to adapt to an environment that is deemed uncooperative or unenthusiastic in a very organic manner [33]. Such optimism has the tendency to take charge of a person by safeguarding their mental health [34]. On a surface level, both hope and optimism could seem very similar; however, both these aspects vary from each other through their distinctive features. Optimists may believe that things will turn out the way they want to, but may not possess the pathways to pursue goals related to what they hope to achieve. Furthermore, optimism focuses broadly on the quality of future outcomes, whereas hope focuses directly on the personal attainment of pursed goals and one’s beliefs of their capability to achieve those goals.

Rand et al. (2020) investigated the role of hope and optimism in college students’ academic performance and subjective well-being [35]. The researchers discovered that college students with high hopes perform better in terms of subjective well-being. It is predicted that having high hope would result in an increase in positive effects and life satisfaction, whereas having high optimism will result in a decrease in negative effects. These results correspond with previous research that revealed a positive association between optimism and life satisfaction, positive effects, and psychological well-being, and a negative relationship between optimism and negative effects [36]. In another study conducted by Chang [37,38,39,40,41], the researcher found that optimism was negatively correlated with depression; meanwhile, it was positively correlated with life satisfaction. This means that the more optimistic an individual is, the lower the depression level. Additionally, the more optimistic an individual is, the more satisfied an individual is with his or her life. Moreover, hope also was found to have the same findings, where researchers have found that it was positively related with life satisfaction and negatively related with depression [42,43]. In another study on hope and optimism, ref. [44] reported similar findings. Both hope and optimism were found to be significant predictors of depression and life satisfaction. However, it was found that hope had a uniquely indirect impact on performance through grade expectancy. Furthermore, a recent study by Genç et al. [21] reported that being hopeful and optimistic are the possible resources to explain how stress from COVID-19 is associated with subjective well-being among college students in Turkey. However, hope plays a more crucial role as a coping mechanism in maintaining the life satisfaction in individuals. This is supported by another study, where [45] found strong evidence that hope has a significantly indirect impact on students’ life satisfaction. It means having high levels of hope could lead to better life satisfaction and well-being than having optimism.

## 2. Method

### 2.1. Participants and Procedure

The participants in this research were 133 private university students in Malaysia. The sample frame selected to conduct this study included young people who are in the emerging adulthood period and who are full time students. G*Power was employed in order to determine the minimum required sample size in terms of statistical power [46]. The model of this study had two predictors. By using G*Power with an effect size of 0.15, alpha of 0.05, and a power of 0.95, the minimum sample size needed was only 107. Thus, with a sample size of 133, our study was large enough and the findings could be confidently reported. The recruitment of participants was conducted via online using a homogeneous convenience sampling method. Firstly, we identified social networking sites such as Facebook, WhatsApp, and LinkedIn. Secondly, the participant that fulfilled the inclusion criteria was approached to complete the self-reported survey. Data collection was held over a two-week period in December 2021 during which 140 responses were gathered. Because of not studying in a private university, seven participants were removed from the dataset; the remaining 133 questionnaires were usable for further analysis, equating to an 95% response rate. Ethical consideration and approval to conduct the study had been obtained from the Scientific and Ethical Review Committee of INTI International University (ref no: INTI/UEC/2021/002).

Of the 133 respondents, 74.4% of them were females and 25.6% were males. In the age category, the largest group of respondents was aged between 18 and 21 years, constituing 57.9% of the total, whereas 48.0% of respondents were from the group aged 22 to 25 years old, followed by group aged between 26 and 29 at 6.0%. In terms of ethnicity, 60.9% of the respondents were Chinese; followed by Malay at 23.3%; Indian at 11.3%, and others at 4.5% of the total. In addition, 78.9% of respondents were single, 18.0% were in a relationship, and 3.0% were married (refer to Table 1).

### 2.2. Measures

A five-section questionnaire was designed in English. Section A consisted of questions about sociodemographic profiles, such as gender, age group, ethnic group, and marital status. The subsequent sections, B to E, encompassed several scales measuring different variables in this study.

Life satisfaction was measured by satisfaction with life scale (SWLS; [47]) using the 7-Point Scale, ranging from 1 (strongly disagree) to 7 (strongly agree). The SWLS was designed to measure global cognitive judgments of one’s life satisfaction. Higher scores represent higher life satisfaction. The Cronbach alpha coefficient of this study was 0.90.

Hope was measured by adult hope scale (AHS; [26]) using the 8-Point Scale ranging from 1 (definitely false) to 8 (definitely true). This scale consists of 12 items with 4 items for each subscale on agency (e.g., “I energetically pursue my goals) and pathways (e.g., “I can think of many ways to get out of a jam”). The remaining 4 items are filler. A mean score was computed where a higher score corresponds to higher levels of hope. The Cronbach alpha for the scales was 0.91 for this study.

Optimism was measured by adapting four items from [48] using the 3-Point Scale, ranging from 1 (lower/lesser/worse) to 3 (higher/greater/better). In order to examine the extent to which students felt optimistic about the future, the following items were used: “Overall, do you think the quality of your life is likely to be higher or lower than your parents’ has been?”; “Overall, do you think your financial well-being in adulthood is likely to be better or worse than your parents’ has been?”; “Overall, do you think your career achievements are likely to be greater or lesser than your parents’ have been?”; and “Overall, do you think your personal relationships in adulthood are likely to be better or worse than your parents’ have been?” The Cronbach alpha for the scales was 0.71 for this study.

Environmental quality was measured by the environment domain of the World Health Organization Quality of Life: Brief Version (WHOQOL-BREF; [49]) using the 5-point Likert Scale. This domain consists of eight items (e.g., “How healthy is your physical environment”; “To what extent do you have the opportunity for leisure activities?”; “How satisfied are you with your access to health services?”). A mean score was computed where a higher score corresponds to higher quality of environment. The Cronbach alpha for the scales was 0.91 for this study.

### 2.3. Data Analysis

This study applies PLS-SEM to gain greater insights into the mediating effect of hope and optimism on the relationship between environmental quality and life satisfaction. We employed PLS-SEM due to the inherent suitability of this approach for exploratory studies [50], which could examine both the measurement and structural models [51], using the SmartPLS 3.0 software package [52]. To perform the mediator analysis, the product of the coefficients approach using bootstrapping has been applied [51].

## 3. Results

This study conducted a two-step process that includes assessment of the measurement and structural models [53].

### 3.1. Assessment of the Measurement Model

The measurement model entails establishing the reliability, convergent validity, and discriminant validity. The framework used in this study consisted of four reflective constructs, namely, environmental quality, hope, optimism, and life satisfaction. In order to establish construct reliability, the outer loadings, composite reliability (CR) and rho_A should be greater than 0.7. Moreover, to establish the convergent validity, the average variance extracted (AVE) should be greater than 0.5 [50]. As shown in Table 2, the result of the measurement model assessment using the criteria has met the required thresholds, therefore indicating that reliability and convergent validity are established for the study model.

To ascertain the discriminant validity of the constructs in this study, we applied the heterotrait-monotrait (HTMT) ratio [53]. Discriminant validity is achieved when the HTMT ratio is less than 0.85 [54]. Table 3 shows the value of HTMT for all constructs is lower than 0.85, indicating that discriminant validity has been achieved for this study model.

### 3.2. Assessment of the Structural Model

In assessing the structural model and hypothesis testing, the collinearity between research variables and R^2^ of the endogenous constructs were examined. *t*-Value and 95% bias-corrected confidence intervals were used to evaluate the sign and significance of the path coefficient [54]. In addition, the indirect effect was examined using *t*-value and 95% bias-corrected confidence intervals to identify and therefore assess the mediation hypothesis [54]. Table 4 shows that the VIF values of all constructs were below 5 (ranged 1.000–1.077) indicating no multi-collinearity issue. The R^2^ values for hope, optimism, and life satisfaction were 0.524, 0.040, and 0.384, respectively, which are considered acceptable for behavioral science studies [54]. Furthermore, the direct effects of environmental quality on hope (*β* = 0.504, *t* = 7.37, *p* < 0.01) and hope on life satisfaction (*β* = 0.589, *t* = 9.42, *p* < 0.01) are significant. However, the direct effect of environmental quality on optimism (*β* = 0.200, *t* = 2.16, *p* > 0.01) and optimism and life satisfaction (*β* = 0.092, *t* = 1.24, *p* > 0.01) are not significant (refer to Figure 1). The results support the indirect effects of environmental quality on life satisfaction through hope (*β* = 0.297, *t* = 4.62, *p* < 0.01), but did not support the indirect effects of environmental quality on life satisfaction through optimism (*β* = 0.018, *t* = 0.92, *p* > 0.01). Therefore, this study shows that by improving hope, environmental quality will increase life satisfaction for private university students.

## 4. Discussion

The present study gives a preliminary understanding into the comprehension of the perplexing idea of the association between perceived environmental quality and life satisfaction by examining the mediating effect of optimism and hope. Results from this study revealed that perceived environmental quality indirectly affects life satisfaction through private university students’ sense of hope, but not optimism. Our results suggested that a higher perceived quality of environment leads to higher hope, which in turn results in higher life satisfaction among university students. True to form, living under isolation for quite a long time worrying about well-being and security, living place, and physical environment affected students. Therefore, comprehending the association between perceived environmental quality and underlying factors may help to investigate the effect of environmental quality on life satisfaction, which is vital for mental health providers to produce effective mental health services amidst pandemics.

Current review results uncovered that hope helped in adapting to environmental quality concerns during the COVID-19 pandemic, but not optimism, in a manner predictable with the study conducted by [35]. Hope has been connected with positive mental well-being results and well-being [55], and hope could very much be vital for college undergraduates and aid them in overseeing climate concerns during pandemics. Investigations into past research have additionally declared that possessing hope is a critical part in channelizing life satisfaction [56]. Moreover, individuals with more noteworthy expectations in the form of hope apply more adaptive coping strategies to oversee unfavorable climate conditions [57], and hope additionally prompts individuals to change their relationship with pessimistic contemplations and feelings by zeroing in on positive energy that works on their capacity to adapt to terrible natural conditions which revolve around life satisfaction. Another clarification may be that confident individuals might assist them with turning out to be proactively occupied with their objective interests, as they would lean towards being positive and useful in the midst of an emergency [58]. Consequently, having a feeling of hope safeguards the well-being of college undergraduates’ and it reduces the pessimistic impacts of adverse environmental conditions, as demonstrated by many investigations [21].

A striking distinction among optimism and hope is that the confidence built is not guaranteed to include pathways and inspirational thinking in the form of motivation [59]. Getting ready to safeguard oneself amidst COVID-19 includes some proportion of individual control [24]. Thus, hope might be more powerful in the midst of the pandemic where people see more noteworthy control of the climate. Hope is moored in convictions around one’s capacities to accomplish objectives (i.e., “I can imagine numerous ways of getting the things in life that are essential to me.”). In circumstances where undergraduates see command or control over an ideal result, hope probably impacts specific expectancies about a specific wanted result (e.g., “In any event, when others get deterred, I realize I can figure out how to tackle the problem.”). Alternately, on the grounds that optimism is moored in convictions broader than oneself, it could be more compelling in circumstances where the activities of oneself are less applicable (i.e., “the nature of my life is probably going to be higher or lower than my folks”). For instance, more prominent positive thinking associated with optimism might work with the conviction that everything will pan out for something good in any event, even when the conditions are not controllable (e.g., “my vocation accomplishments are probably going to be more noteworthy than my folks’ have been”). This hopeful anticipation might prompt more prominent persistence and emotional well-being while looking out for a wild result. During the beginning phase of the pandemic, undergraduates’ thought process of being an impermanent bother ended up being a better approach for life. Any optimistic people would imagine that we will be back to our ordinary lives soon, just for the present circumstance to turn out to be better than just being frustrated with the pandemic being delayed and remaining significantly longer than whatever they had anticipated. Notwithstanding, a hopeful individual will really make a move to accomplish what they need to adjust to the new standard. Hopeful individuals find it more challenging to achieve life fulfillment since they endeavor to keep away from adverse results and endeavor towards positive and desired outcomes [27].

### Limitations, Implications, and Conclusions

The current study has made several implications to theoretical and practical development. Our study has provided an initial insight into the mediating effect of hope and optimism between environmental quality and life satisfaction among private university students during COVID-19 in Malaysia. The findings of our research also contribute to the theory of hope [26]. The theory was formulated by Snyder [26,29] and the researcher noted the importance of taking cultural and societal background into consideration in the discussion of applications. Our study was carried out in Malaysia, a country where collectivism is widely practiced, and this has influenced the factors mentioned above. Thus, the findings of our study contribute not only additional empirical data to the theory but also help to view the theory from the pandemic point of view. Moreover, our study also contributes more empirical data and evidence to the topic of life satisfaction during COVID-19 in Malaysia.

Secondly, this study contributes to continuous practical implications on life satisfaction, which is an essential ingredient in subjective well-being [60]. Consequently, current research hopes to encourage mental health providers to implement hope-focused intervention in their practices to help in reducing students’ life stressors. Students need to be trained to contemplate their own capacity to carry objectives to completion, including the capacity to create courses to arrive at objectives and the inspiration to utilize those courses. Hope-focused intervention may also be incorporated into other forms of psychotherapy such as cognitive behaviour therapy, since hope has been shown as one of the protective factors against psychological issues [22]. Additionally, counsellors may consider integrating hope into their counseling sessions. Students should be trained on utilizing the power of hope when they are facing an environmental crisis or having problems maintaining satisfaction in life.

This study does, however, have a number of limitations. The mediation analysis does not determine the cause and effect relationship; thus, a longitudinal study with manipulation of variables could be carried out in the future. In addition, the sample comprises only private university students. Future studies should recruit samples from different settings, including school children, in order to improve generalizability of the results. International studies are warranted in determining the role of the school environment on the transmission of the COVID-19 virus and the indirect influence on the health of students in general [61].

## Figures and Tables

**Figure 1 ijerph-19-07661-f001:**
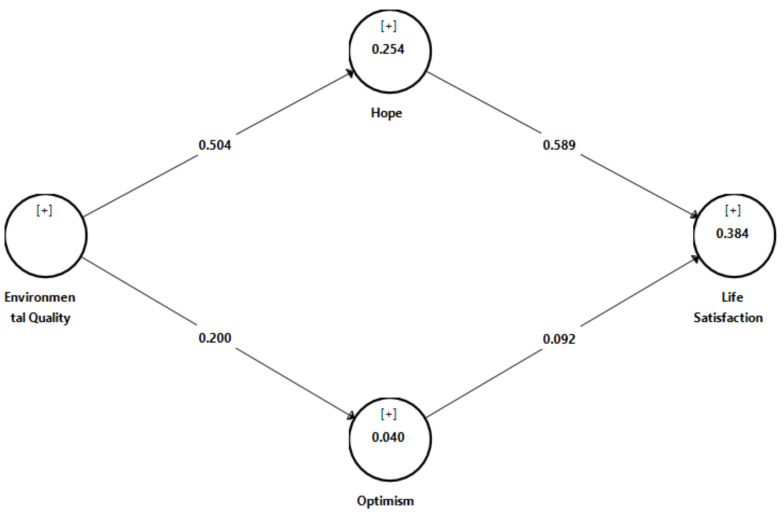
Results of assessment of structural model.

**Table 1 ijerph-19-07661-t001:** Demographic Profile of Respondents (*N* = 133).

Variables		Frequency	%
Gender	Male	34	25.6
	Female	99	74.4
Age	18–21 years old	77	57.9
	22–25 years old	48	36.1
	26–29 years old	8	6.0
Ethnicity	Malay	31	23.3
	Chinese	81	60.9
	Indian	15	11.3
	Others	6	4.5
Marital Status	Single	105	78.9
	In a relationship	24	18.0
	Married	4	3.0

**Table 2 ijerph-19-07661-t002:** Results of measurement model assessment.

Construct	Loadings	CR	rho_A	AVE
Environmental Quality		0.864	0.816	0.517
EQ1	0.600			
EQ2	0.728			
EQ3	0.805			
EQ4	0.736			
EQ5	0.662			
EQ6	0.766			
Hope		0.924	0.923	0.607
H1	0.674			
H2	0.864			
H3	0.807			
H4	0.866			
H5	0.666			
H6	0.799			
H7	0.774			
H8	0.754			
Optimism		0.814	0.728	0.594
OP1	0.846			
OP2	0.694			
OP3	0.766			
Life Satisfaction		0.927	0.919	0.719
LS1	0.878			
LS2	0.848			
LS3	0.91			
LS4	0.860			
LS5	0.734			

**Table 3 ijerph-19-07661-t003:** Discriminant validity through HTMT_0.85_.

	Environmental Quality	Hope	Life Satisfaction	Optimism
Environmental Quality				
Hope	0.581			
Life Satisfaction	0.663	0.645		
Optimism	0.239	0.323	0.27	

**Table 4 ijerph-19-07661-t004:** Results of hypothesis testing.

Direct/Indirect Effect	Path Coefficient	*t*-Value	95% Bias-Corrected Confidence Interval	Supported
EQ → Hope	0.504	7.37	[0.357, 0.620]	Yes
EQ → Optimism	0.200	2.16	[−0.170, 0.316]	No
Hope → LS	0.589	9.42	[0.448, 0.689]	Yes
Optimism → LS	0.092	1.24	[−0.062, 0.244]	No
EQ → Hope → LS	0.297	4.62	[0.167, 0.412]	Yes
EQ → Optimism → LS	0.018	0.92	[−0.012, 0.067]	No

## Data Availability

Data can be made available upon reasonable request.

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
