# Peer review of "The Unique Role of Hope and Optimism in the Relationship between Environmental Quality and Life Satisfaction during COVID-19 Pandemic"

_ijerph, 2022, doi:10.3390/ijerph19137661_

Round 1

Reviewer 1 Report

Dear Authors,

Thank you for your manuscript and please find my remarks and comments.
The presented paper deals with an interesting topic of life satisfaction taking into consideration Covid-19 pandemic.
Below you can find several remarks which might be taken into consideration:

General comments

If I am correct, you have assumed that one’s life satisfaction is related to the environmental conditions. But – the impact of environmental conditions on the life satisfaction is (additionally) “filtered” or “interfered” by hope and/or by optimism. 
In the section “3.2. Assessment of structural model” you have stated that (line 224-230)
“The results support the indirect effects of environmental quality on life satisfaction through hope (β = 0.297, t = 4.62, p < 0.01), but did not support the indirect effects of environmental quality on life satisfaction through optimism (β = 0.018, t = 0.92, 228 p > 0.01)” => what is directly associated with the study goal.
But, then you put a comment (line 240 – 242) “Our results suggested that higher perceived quality of environmental leads to a have higher hope, which in turn have higher life satisfaction among university students”.
My question is – what is the cause, and what is the effect in your study? I just want you to consider this topic, and maybe present in the way, which might more easy to follow to the audience. We can ask for instance: does higher hope impact my perception of environmental quality – and as a consequence, make me life satisfaction higher?

More specific comments / questions
1/ Line 80
“findings of this study could help mental health providers to provide practical skills to” => please rephrase 
2/ Line 119 
“Furthermore, a recent study by [19] reported” => I guess you forgot to put the name oh the Author you refer to.
3/ Line 159-160 
you stated “…using the 5-Point Scale, ranging from 1 (strongly disagree) to 7 (strongly agree)”. => I am not sure whether 5 or 7 point scale was used. 
4/ Line 162
“…study was .90.” => please change to 0.90
5/ I miss the questionnaire in the appendix – it would be much easier to understand your approach and the whole paper.

Good luck,
Reviewer

Author Response

Dear Examiner,

We are grateful for your consideration of this manuscript, and we also very much appreciate your suggestions, which have been very helpful in improving the manuscript. All the comments we received on this manuscript have been taken into account in improving the quality.

Reviewer 2 Report

It was a pleasant experience reading this paper.

It sheds a light on some aspects poorly investigated during these years of pandemic studies.

Nonetheless, while it is interesting the role attributable to positive feelings for the quality of life for young university students, one should not forget that schools was one of the factors (besides others) that was demonstrated at the basis of the diffusion of this virus.

Not to neglect this aspect, I would expect that this should be recognized in the Introduction, or in the Discussion, of the paper by citing some relevant works.

I would suggest the two following one, at least to mention and to cite, nonetheless the authors could further add someone similar study conducted in their country of interest.

Done these corrections, the paper can become publishable.

1- 

Casini L, et al Reopening Italy’s schools in September 2020: a Bayesian estimation of the change in the growth rate of new SARS-CoV-2 cases     Flasche S, et al.

Author Response

(The authors gave the same response as above.)

Reviewer 3 Report

Dear Authors,

Thanks for giving me the chance to read this manuscript, “The Unique Role of Hope and Optimism in the Relationship between Environmental Quality and Life-Satisfaction during COVID-19 Pandemic”. The current paper tries to analyze s the preliminary understanding of the intervening role of hope and optimism in the relationship between environmental quality and life satisfaction among private university students in Malaysia.

This is an interesting topic in the field of psychological health during the COVID-19 Pandemic. However, there are significant issues in the current manuscript that should be carefully addressed to be further considered.

  1. Literature review

  • The literature review did not thoughtfully discuss the relationship between the four factors, quality, optimism, hope, and life satisfaction.

  1. Design and Method

  • The proposed model is much too simple which must be further developed and explored. Please consider adding more factors to have a comprehensive analysis of related factors.
  • Why choose a university sample? just for convenience?
  • No valid test?
  • No reliability test?

  1. Discussion

  • The discussion part should be rewritten where the theoretical and practical contributions should be emphasized.

  1. Language

  • The current version needs in-depth proofreading. Many typos and grammar issues have been found in the manuscript, e.g., the word “suggesting” in Line 56.

To sum up, I personally like this paper. However, the problems should be addressed in order to be further considered. Hope these suggestions help.

Author Response

(The authors gave the same response as above.)

Round 2

Reviewer 2 Report

The situation is not very different from before: My comments have been only partially addressed. My evaluation is the same: currently not yet publishable, take my previous comments and implenemt them ALL!!!

Reviewer 3 Report

Dear Authors, 

Although this research is very interesting, it seems that the authors do not rework the empirical part which is the main concern from my perspective.

I would suggest declining the current version and encouraging you to resubmit.

Authors are highly recommended to consider the previous comments and expand the current model.

Best,
